# QTL Verification and Candidate Gene Screening of Fiber Quality and Lint Percentage in the Secondary Segregating Population of *Gossypium hirsutum*

**DOI:** 10.3390/plants12213737

**Published:** 2023-10-31

**Authors:** Ruixian Liu, Minghui Zhu, Yongqiang Shi, Junwen Li, Juwu Gong, Xianghui Xiao, Quanjia Chen, Youlu Yuan, Wankui Gong

**Affiliations:** 1Engineering Research Centre of Cotton, Ministry of Education, College of Agriculture, Xinjiang Agricultural University, 311 Nongda East Road, Urumqi 830052, China; liuruixian1223@163.com; 2National Key Laboratory of Cotton Bio-Breeding and Integrated Utilization, Institute of Cotton Research of Chinese Academy of Agricultural Sciences, Anyang 455000, Chinagongjuwu@caas.cn (J.G.); xiaoxianghui4953@163.com (X.X.); 3Agricultural Technology Extension Center of Kashi District, Kashi 844000, China; 13779727908@163.com; 4Zhengzhou Research Base, National Key Laboratory of Cotton Bio-Breeding and Integrated Utilization, Zhengzhou University, Zhengzhou 450001, China

**Keywords:** *Gossypium hirsutum*, secondary segregating population, fiber quality, lint percentage, QTL

## Abstract

Fiber quality traits, especially fiber strength, length, and micronaire (FS, FL, and FM), have been recognized as critical fiber attributes in the textile industry, while the lint percentage (LP) was an important indicator to evaluate the cotton lint yield. So far, the genetic mechanism behind the formation of these traits is still unclear. Quantitative trait loci (QTL) identification and candidate gene validation provide an effective methodology to uncover the genetic and molecular basis of FL, FS, FM, and LP. A previous study identified three important QTL/QTL cluster loci, harboring at least one of the above traits on chromosomes A01, A07, and D12 via a recombinant inbred line (RIL) population derived from a cross of Lumianyan28 (L28) × Xinluzao24 (X24). A secondary segregating population (F_2_) was developed from a cross between L28 and an RIL, RIL40 (L28 × RIL40). Based on the population, genetic linkage maps of the previous QTL cluster intervals on A01 (6.70–10.15 Mb), A07 (85.48–93.43 Mb), and D12 (0.40–1.43 Mb) were constructed, which span 12.25, 15.90, and 5.56 cM, with 2, 14, and 4 simple sequence repeat (SSR) and insertion/deletion (Indel) markers, respectively. QTLs of FL, FS, FM, and LP on these three intervals were verified by composite interval mapping (CIM) using WinQTL Cartographer 2.5 software via phenotyping of F_2_ and its derived F_2:3_ populations. The results validated the previous primary QTL identification of FL, FS, FM, and LP. Analysis of the RNA-seq data of the developing fibers of L28 and RIL40 at 10, 20, and 30 days post anthesis (DPA) identified seven differentially expressed genes (DEGs) as potential candidate genes. qRT-PCR verified that five of them were consistent with the RNA-seq result. These genes may be involved in regulating fiber development, leading to the formation of FL, FS, FM, and LP. This study provides an experimental foundation for further exploration of these functional genes to dissect the genetic mechanism of cotton fiber development.

## 1. Introduction

Cotton fiber is an important raw material in industrial production [1]. Owing to its high yield and wide adaptability, allotetraploid upland cotton is widely planted around the world, accounting for over 95% of the world’s total cotton planting areas. However, its fiber quality is less attractive than sea-island cotton [1,2]. With the continuous upgrading and innovation of textile technology, the requirements for the quality and quantity of cotton fibers are constantly increasing [3]. Cotton fiber quality is mainly composed of fiber length (FL), fiber strength (FS), fiber micronaire (FM), fiber uniformity (FU), and fiber elongation (FE) [4]. The lint percentage (LP) is a main indicator for evaluating cotton fiber yield [5]. The cost-effectiveness of raw cotton fiber production requires high fiber quality that meets the technical requirements of the textile industry and a high yield potential that reimburses the agricultural costs. Therefore, effectively improving the fiber quality and yield of upland cotton remains one of the main goals of current cotton breeding projects [6].

Fiber quality and yield traits are complex traits controlled by multiple minor quantitative trait loci (QTLs)/genes and are easily affected by environments [7,8]. Conventional breeding methods have made great contributions to the improvement of both fiber quality and yield. However, the limitations in the further simultaneous improvement of the two are becoming increasingly prominent due to the negative correlations between them, which become obvious under the current high-level breeding conditions [9]. QTL mapping provides a powerful approach for cotton breeders to improve the fiber quality and yield traits of upland cotton via marker-assisted selection (MAS) [9,10]. With the continuous improvement in the reference genome and the large-scale sequencing of a large number of bi-parental segregating and natural populations, a large number of QTLs and QTNs of fiber quality, yield traits, and biotic stresses, etc., have been identified and associated, providing important resources for cotton breeding programs via MAS [4,10,11]. However, the practical value of these QTLs in MAS lies in their ability to pyramid the favorable alleles into newly developed cultivars. Furthermore, a large number of QTLs still need to be validated or fine-mapped for practical MAS applications or functional gene studies [12,13,14]. With the improvement in genome and genome annotation information and the application of transcriptome technology, researchers will be able to identify candidate genes based on the annotation information and transcriptional expression profiles of genes within the QTL interval and explore gene functional research [3,12]. Fang et al. [15] constructed an F_2_ population (CCRI35 × Yumian1) containing 2484 individual plants to fine map an FS QTL, qFS07.1, to a DNA fragment of 62.6 kb (0.17 cM) containing four annotated genes. Through qRT-PCR and sequence comparison analysis, a leucine-rich repetitive protein kinase (LRR RLK) family protein-coding gene, *Gh_A07G1749*, was identified as its candidate gene. Islam et al. [3] used 27 SNP markers to finely map multiple QTLs, namely qFBS-c3, qSFI-c14, qUHML-c14, and qUHML-c24, in an intraspecific F_3_ population (MD90ne × MD52ne) of upland cotton and accurately mapped them to the physical intervals of 4.4 Mb, 1.8 Mb, and 3.7 Mb in the reference genome. The receptor kinase pathway gene was identified as a candidate gene responsible for FS and FL based on the differential expression profiles and the amino acid mutation analysis of the gene between the two parents. Zang et al. [12] constructed four F_2_ populations containing 1864 individuals through backcrossing four recombinant inbred lines (RILs) derived from the cross of Prema×86-1, RIL43, RIL98, RIL120, and RIL168, with 86-1, respectively. qFS-D3-1 was finely mapped to a fragment of 0.93 Mb (1.14 cM), which contained 23 annotated genes. Based on the transcriptional expression profiles of these genes during fiber development and gene sequencing, an allele with a 6 bp (GCCTCC) deletion of *GhUBX* (*GH_D03G0985*) gene was identified to be responsible for higher FS in Prema. *GhUBX* regulates fiber helix growth by degrading GhSPL1 in fiber cells through the ubiquitin 26S–proteasome pathway, leading to an increase in the number of cotton fiber helices and eventually improving fiber strength. Liu et al. [14] finely mapped qSI-A07-1 to 17.45 kb via an F_2_ population and identified an allele with a deletion of 845 bp in the intron region of its candidate gene *GH_A07G2179* (*GhSI7*; transcriptional regulator STERILE APETALA) responsible for regulating seed size. However, how these candidate genes altogether regulate cotton fiber quality development yield formation still keeps elusive. In the current climate changing conditions, research on the genetic regulation mechanisms of fiber yield and quality formation has become increasingly important, which may impose a great impact on the future development of both cotton cultivation and textile industry.

In a previous study, three QTL clusters were identified via a RIL population derived from an intraspecific upland cotton cross of Lumianyan28 and Xinluzao24 (L28 × X24) [16]. The clusters mainly consisted of major-effect QTLs for their target traits of FS, FL, FM, and LP (Table 1). These loci may have potential implications for future variety improvement and for dissecting the formation mechanism of target traits. In this study, a secondary segregating F_2_ population was constructed via a cross of L28 and RIL40, which was an excellent fiber quality RIL-derived L28 × X24. An F_2:3_ population derived from F_2_ was also constructed. Both F_2_ and F_2:3_ populations were applied to verify the genetic effect of the QTLs in the above three clusters, and the potential candidate genes were identified via analysis of the differentially expressed genes (DEGs) within the QTL interval based on an RNA-seq strategy. The results revealed that these QTLs will be of great significance in future breeding projects, and further dissection of the candidate genes will be beneficial to understanding their acting mechanism in cotton fiber development.

## 2. Results and Analysis

### 2.1. Phenotypic Statistics of Fiber Quality and Yield Traits of the Experimental Materials

Basic phenotypic statistics describing the FL, FS, FM, and LP of the parental lines, and the F_2_ and F_2:3_ populations are presented in Table 2. The RIL40 had higher FL and FS phenotypic values and lower FM and LP phenotypic values than those of L28, which were the same as those observed in the previous study (Table 2) [16]. The t test revealed that the differences in these traits between RIL40 and L28 reached at least significance at *p* ≤ 0.05 (Table 2). The skewness and kurtosis evaluations showed that the phenotypes of all the target traits fit a normal distribution in both the F_2_ and F_2:3_ populations and that the F_2:3_ phenotypic values could represent the variations and distributions of those of F_2_ (Figure 1). Correlation analysis revealed that the phenotypic performances of these target traits were significantly positively correlated between the F_2:3_ and F_2_ generations. However, within each generation, the trait pairs of FL-FS and FM-LP were significantly positively correlated, while those of FL-FM/LP and FS- FM/LP were significantly negatively correlated in both the F_2:3_ and F_2_ (except FM-FS in F_2_ and LP-FS in F_2:3_) (Table 3).

### 2.2. Linkage Map Construction and QTL Mapping of the Target Loci

All polymorphic markers were used to genotype the 1961 individual plants of the F_2_ population (Appendix A). Three genetic linkage groups were finally constructed for the target loci on chromosomes A01, A07, and D12 (Table 1), each of which contained 2, 14, and 4 molecular markers, spanning 12.25, 15.90, and 5.56 cM, respectively (Figure 2). The linkage groups of the previous study, the physical positions of the markers, and the linkage groups of the current study are presented in Figure 2a–c. The results revealed that the linkage groups of the current study were consistent with those of the previous study.

QTL analysis of these three linkage groups on chromosomes A01, A07, and D12 of 1961 F_2_ individuals and 356 F_2:3_ lines revealed that the QTLs identified in the linkage groups were consistent with the ones identified at the same loci in the previous study. Namely, in the linkage group on A01, an FS QTL qFS-A01-1 was identified, which spanned a physical interval of 0.27 Mb (7.73–8.00 Mb in the physical map). In the linkage group on A07, four QTLs, including qFL-A07-1, qFS-A07-1, qFM-A07-1, and qLP-A07-1 for each trait, respectively, were identified, which spanned a physical interval of 2.24 Mb (88.95–91.29 Mb in the physical map). In the linkage group on D12, an FL QTL qFL-D12-1 was identified, which spanned a physical interval of 0.11 Mb (0.48–0.59 Mb in the physical map) (Table 4, Figure 2c).

### 2.3. Screening and Analysis of DEGs from Genes within the QTL Intervals

In the QTL intervals, a total of seven DEGs, including one, five, and one on A01, A07, and D12, respectively, were identified, which were differentially expressed between the parental lines, L28 and RIL40, in the corresponding stages of fiber development via RNA-seq analysis (Table 5). In these DEGs, two, *GH_A07G2180* and *GH_A07G2209*, were highly expressed at 10 DPA during fiber development, and their expression gradually decreased after 20 DPA; one gene, *GH_A07G2203*, was highly expressed at 20 DPA during fiber development; three genes, *GH_A07G2222*, *GH_A07G2247*, and *GH_D12G0031*, were highly expressed at 30 DPA during fiber development; one gene, *GH_A01G0633*, was similarly expressed at 10, 20, and 30 DPA during fiber development (Figure 2d).

Genes in the QTL intervals of A01, A07, and D12 were screened via their FPKM values in developing fibers at 10, 15, 20, and 25 DPA of TM-1, which were fetched from the cotton functional genomic database (CottonFGD: https://cottonfgd.net/, accessed on 30 October 2022). The genes that had a mean FPKM value >0.5 were regarded as expression genes. A total of 79 genes from the interval on the A01 chromosome, 133 genes on the A07 chromosome, and 49 genes on the D12 chromosome were identified to have a dynamic expression during fiber development, forming six, six, and four distinct expression clusters, respectively (Figure 3). The results revealed that the DEG *GH_A01G0633* was identified in expression cluster 1 of the interval on chromosome A01, which exhibited a steadily decreasing expression trend (Figure 3). The DEGs *GH_A07G2180*, *GH_A07G2203*, *GH_A07G2209*, *GH_A07G2222*, and *GH_A07G2247* were identified in expression cluster 4, 6, 2, 5, and 1 on chromosome A07, respectively (Figure 3). The DEG *GH_D12G0031* was identified in expression cluster 2 of the interval on chromosome D12, which showed a high expression at 15 DPA and then went down in slightly different styles (Figure 3). These results indicated the DEGs’ involvement in the fiber development of cotton plants.

qRT-PCR verification using fiber samples of L28 and X24 at 5, 10, 15, 20, 25, and 30 DPA confirmed the differential expression of the seven DEGs between L28 and X24. It also revealed that five DEGs, *GH_A01G0633*, *GH_A07G2180*, *GH_A07G2222*, *GH_A07G2247*, and *GH_D12G0031*, were consistent with the results of the RNA-seq analysis, while two DEGs, *GH_A07G2203* and *GH_A07G2209*, were inconsistent (Figure 4). The consistent results of the RNA-seq, TM-1 gene expression data analysis, and qRT-PCR indicated the important roles of these five DEGs in cotton fiber development.

## 3. Materials and Methods

### 3.1. Plant Materials and Phenotypic Measurement

In a previous study, a *G. hirsutum* RIL population was established using a cross between two upland cultivars Lumianyan28 (L28) and Xinluzao24 (X24), L28 × X24 [16]. L28 is a conventional cotton cultivar showing high yield potential, while X24 is a cultivar possessing high-quality fibers. The QTLs of the fiber quality and yield traits were identified, of which three cluster loci were selected for further analysis in the current study (Table 1). For this purpose, RIL40, a line of the L28 × X24 RIL population [16], which had the favorable alleles from X24, was selected to cross L28 to develop a secondary F_2_ population. The developmental procedures were as follows: In the 2017 winter growing season, a cross of L28 × RIL40 was made at the experimental station of the Institute of Cotton Research in Sanya (Hainan province, China), the F_1_ seeds were harvested and then planted in Anyang (Henan province, China) in the 2018 summer growing season. The F_1_ plants were self-pollinated to obtain F_2_ seeds in the 2018 summer growing season in Anyang. The F_2_ population and parental lines, L28, RIL40, and X24, were planted in the 2019 summer growing season in Anyang. The F_2_ were harvested per plant to form F_2:3_ seeds, and an F_2:3_ population was planted in two replications in a completely randomized block design in the 2020 summer growing season in Anyang. All naturally opened bolls were hand harvested per plant from the F_2_ plants, and per line from the F_2:3_ and the parental lines, respectively. The seed cotton of each sample was weighed and then ginned. The LP of each sample was evaluated, and the fiber quality traits were evaluated using the HVI (High Volume Instrument) system at the Institute of Cotton, Hebei Academy of Agriculture and Forestry Sciences (Shijiazhuang, China). The fiber quality traits include the FL (mm), FS (cN/tex), and FM. The descriptive statistics and correlation analysis of phenotype data were calculated using SPSS 21 software. Bar plots were created using OriginPro 2021 software.

### 3.2. Maker Development for Genotyping of the Secondary Population

Total genomic DNA was extracted from young leaves of F_2_ individuals and the parental lines, L28 and RIL40, using a modified CTAB method [17]. Three simple sequence repeat (SSR) markers, SWU2707, DPL0852, and DPL0757, reported in previous studies [16] were directly used to genotype the F_2_ population (Appendix A). De novo-designed molecular markers, SSR and Indel, were based on the TM-1 reference genome [18] within or adjacent to the physical intervals of the three target loci (Table 1). Each primer pair was designed in a ±200 bp sequence interval. All distinctive and unambiguous polymorphic markers between L28 and RIL40 were used to genotype the F_2_ populations. The marker loci were named following their primer names. The genotyping results of the F_2:3_ population were deduced from the results of the F_2_.

### 3.3. QTL Mapping

The genetic linkage maps of the three target regions were constructed using JoinMap 4.0 software [19]. The conversion of the recombination frequencies to map distances (cM) used the Kosambi function [20]. The QTLs were identified by composite interval mapping (CIM) using WinQTL Cartographer 2.5 software [21]. Linkage groups and QTL distribution on the map were visualized using MapChart 2.2 software [22].

### 3.4. Candidate Gene Screening Based on DEG Analysis between L28 and RIL40

Total RNA was extracted from developing fiber samples of L28 and RIL40 at 10, 20, and 30 DPA, using TRIzol reagent (Tiangen, Beijing, China). Three biological replicates were performed for each sample. The raw data of the Illumina NovaSeq6000 sequencing platform had adaptor trimming, low-quality, and short reads processing with Fastp (v.0.20.0) [23] to obtain clean data. Quality control of the clean data was performed using Fastqc (v.0.11.5) [24]. RNA-seq data analysis was performed using BMKCloud (www.biocloud.net, accessed on 24 May 2023). To obtain the location information of clean reads on the reference genome, the clean reads were aligned to and compared with the *G*. *hirsutum* (TM-1_V2.1) reference genome [18] using Hisat2 (v2.0.4) [25] and SAMtools [26]. Then, the mapped reads were reassembled into a transcriptome by StringTie (v2.2.1) [27] based on the *G. hirsutum* (TM-1_V2.1) reference genome [18]. The fragments per kilobase of transcript per million fragments of mapped reads (FPKM) values [28] of all genes were normalized using StringTie (v2.2.1) [27] software, which was used to evaluate the gene expression levels in the fiber developmental stages. The differentially expressed genes (DEGs) between L28 and RIL40 in the same development stage were analyzed using the DESeq2 (v1.30.1) [29] of the R package based on the criteria of |Fold Change| ≥ 2.0 and false discovery rate (FDR) < 0.01.

The FPKM values of the genes of TM-1 at 10, 15, 20, and 25 DPA (days post anthesis) were downloaded from the cotton omics data platform COTTONOMICS (http://cotton.zju.edu.cn/, accessed on 13 October 2022) [30]. Gene expression profile clustering in the QTL cluster intervals was performed based on the FPKM values in the developing fibers of TM-1 using the Mfuzz [31] software (v2.29) in the R package in Hiplot Pro (https://hiplot.com.cn/, accessed on 2 May 2023).

### 3.5. qRT-PCR Experiment

The expression of the candidate genes in the parents of the secondary population was again verified based on the RNA of fibers in different development periods (5, 10, 15, 20, 25, and 30 PDA) of the parents through a qRT-PCR experiment to validate the potential function of candidate genes in different fiber developments. The total RNA at each period of fiber development was isolated by the above and was converted to cDNA using the HiScript III RT superMix for qPCR (+gDNA wiper) reverse transcription (R323-01AA, Vazyme, Biotech, Nanjing, China). The qRT-PCR analyses were performed on an Applied Biosystems 7500 fast real-time PCR system (ABI) utilizing the chamQ universal SYBR qPCR master mix (Q711-02-AA, Vazyme, Biotech, Nanjing, China). The relative expression level of genes was calculated with the 2^−ΔΔCT^ method [32]. All primer sequences for the qRT-PCR analysis are listed in Appendix A. The actin gene was used as a reference gene.

## 4. Discussion

### 4.1. MAS Strategy in Breeding Practice

The screening of functional markers is the key step in MAS-based breeding projects. However, QTLs in previous studies were usually identified via the following key steps, i.e., (a) constructing a linkage segregation population, temporary or permanent; (b) genotyping the individuals of that population using an appropriate set of markers or marker collections; (c) phenotyping the individuals of the population in a certain environment; and (d) using the proper software to calculate the correlation between the genotype and phenotype. If an allele of a locus is significantly correlated to the expression of a trait, then it is thought that a QTL has been identified at this locus. Therefore, the QTLs are usually genetic background dependent and environment tagged. Obtaining effective QTLs that are not constrained by the genetic background and environment and using their markers in future practical breeding practices via MAS still remains of particular interest to researchers. Various studies have tackled this issue via several strategies [33,34]. In a previous study, QTLs of FL, FS, FM, and LP were identified through linkage analysis [16]. To validate the effectiveness of these QTLs, the current study selected three loci consisting of these QTLs or QTL clusters on chromosomes A01, A07, and D12 and used secondary segregation populations, including F_2_ and its derived F_2:3_, to validate them. The result positively confirmed the effectiveness of QTL selection during early generations after hybridization. However, we also noticed that, compared to the previous results [16], the phenotypic variation rates of the target traits explained by the loci of A01 and D12 in the F_2_ or F_2:3_ generations were relatively low (Table 4). The dominant effect is a common phenomenon in QTL identification in temporary populations and it is not always in the same direction as the additive effect [35,36,37]. The reason might be the opposite direction of the dominant effects of the same locus against additive effects. As the ultimate goal in a breeding project is to pyramid the loci of additive effects, except for the utilization of heterosis, the presence of such dominant effects does not affect the effectiveness of MAS in early generations. Moreover, the physical intervals of the three loci in this study were consistent with those of previous reports [14,33,34,38,39,40,41,42]. The results of this study indicate that the target traits, FL, FS, FM, and LP, are interrelated, as the QTLs of these traits are co-located in specific genomic regions. These loci could be effectively used for further research on the formation fiber quality and yield traits, as well as for future breeding projects via MAS, which necessitates an integrative consideration of the QTL compositions in the clusters and the distribution of QTL loci of each trait, especially the ones for the comprehensive improvement of multiple traits.

### 4.2. Function Validations of Candidate Genes

In this study, of the three QTL cluster locations, both the DEG analysis and qRT-PCR results demonstrated that *GH_A01G0633* and *GH_D12G0031* were candidate gene for QTL clusters in chromosomes A01 and D12, respectively. CBSX5 is a member of the CBS domain-containing protein family (PF00571), which might be involved in cell wall synthesis. Studies showed that the promoter region of the *CBSX* gene family members contained numerous stresses and phytohormone-responsive elements, indicating their involvement in regulating various stress responses and plant growth development via the plant hormone pathway in plants [43,44,45,46,47]. In *Arabidopsis*, it was demonstrated that *CBSX* regulates H_2_O_2_ levels and lignin polymerization [45], as well as secondary cell wall thickening of the endothecium during anther dehiscence [46], reactive oxygen species (ROS), and lignin deposition [44]. Previous studies revealed that, in cotton, genes specifically expressed under stress were also specifically expressed during fiber development. In cotton, a study indicated that *GhCBS* genes were regulated under abiotic stress and hormonal treatments and in ovule and fiber development [48], indicating their significant impact on fiber development. *HT1* (*GH_D12G0031*) is a member of the protein kinase superfamily (PF07714), which is involved in protein serine/threonine kinase activity and protein kinase activity. An early study identified that *HT1* was important for the regulation of stomatal movements in response to CO_2_ [49]. Later research revealed that it also regulated red light-induced stomatal opening [50], which is a strong indicator of the water-use efficiency of a plant [51]. Our results suggested that HT1 also played an important role in cotton fiber development; however, its mechanism of action still needs further clarification.

*GH_A07G2222* in the interval of the QTL cluster on chromosome 7 was annotated possibly as BRASSINOSTEROID-INSENSITIVE1 (BRI1)-associated kinase 1 (BAK1)-interacting receptor-like kinase 1 (*BIR1*), a small leucine-rich repeat receptor-like kinase. In Arabidopsis, BIR1 was demonstrated to negatively regulate cell death pathways in plant BAK1-mediated pathogen-triggered immunity (PTI) signaling [52,53]. The functioning of BIR1 is partially dependent on the salicylate (SA)-dependent resistance (R) protein pathway [54]. In higher plants, BAK1 participates in multiple developmental processes through the brassinosteroid (BR) signaling pathway [55,56], including degrading H_2_O_2_ via activation of CAT activity and the development of phloem vascular tissues [57]. As a receptor, BAK1 forms ligand-induced complexes with different LRR-RLKs and flagellin-sensitive 2 in the regulation of vascular development and immune responses via the hormone pathway in plants [57,58], including ABA signaling [59]. *GH_A07G2247* was annotated to encode a member of the glycosyl hydrolase family 17 proteins (GHL17), which hydrolyzes 1,3-b-glucan polysaccharides found in the cell wall matrix [60]. *GHL17* is demonstrated to be involved in physiologically important processes in plants, including responses to biotic and abiotic stresses [61,62,63], defense against herbivores, and activation of phytohormones, lignification, and cell wall remodeling [64,65]. The regulatory mechanisms of these candidate genes in fiber development still need validation. However, previous studies have shown that genes specifically expressed during fiber development are also specifically expressed under stresses [66,67], inferring that cotton fiber development and the stress responses of cotton plants may involve the same metabolic pathways.

In addition, in this QTL cluster interval, another two genes, *GH_A07G2203* and *GH_A07G2209*, were also identified to have specific expression in fiber development. *GH_A07G2203* was annotated as *CONSTANS-like 9* (*COL9*), which is a transcription factor involved in regulating plant flowering and responses to abiotic stress [68,69]. In rice, *OsCOL9* interacted with *OsRACK1* and enhanced rice blast resistance through salicylic acid (SA) and ethylene (ET) signaling pathways [68]; it also regulated the grain number formation of the main panicle in rice plants [69]. In cotton, a study revealed that *COL9* was involved in affecting flowering and the response to drought and salt stresses [70]. *GH_A07G2209* was identified as *RAB GTPase homolog B1C* (*RABBIC*), which is part of the Ras superfamily of small GTPases. RAB GTPases regulate vesicle formation, actin- and tubulin-dependent vesicle movement, and membrane fusion [71]. The RAB GTPase family shares major trafficking elements related to the cell wall modification in ripe fruit, involving the trafficking of cell wall polymers and enzymes between cellular compartments in plants [72,73]. These two candidate genes may also have a possible role in fiber development. In summary, these genes are likely potential candidate genes regulating cotton fiber development, which will be validated in future work.

## 5. Conclusions

In this study, based on the primary QTL identification results via a RIL population developed from L28 × X24, three important QTL or QTL cluster intervals on chromosomes A01, A07, and D12 were selected to conduct further dissection. An F_2_ and its derived F_2:3_ populations were developed from a cross of L28 × RIL40. Genetic linkage maps of these three intervals were constructed using molecule markers (SSR and Indel) using the F_2_ secondary segregating population. The QTLs of fiber quality traits, FL, FS, and FM, and the yield trait LP on these three corresponding intervals were verified. Five DEGs were identified to have important roles in fiber development as potential candidate genes via RNA-seq strategy, TM-1 gene expression data analysis, and qRT-PCT verification. This study provides an experimental foundation for further exploration of these functional genes to dissect the genetic mechanism of cotton fiber development.

## Figures and Tables

**Figure 1 plants-12-03737-f001:**
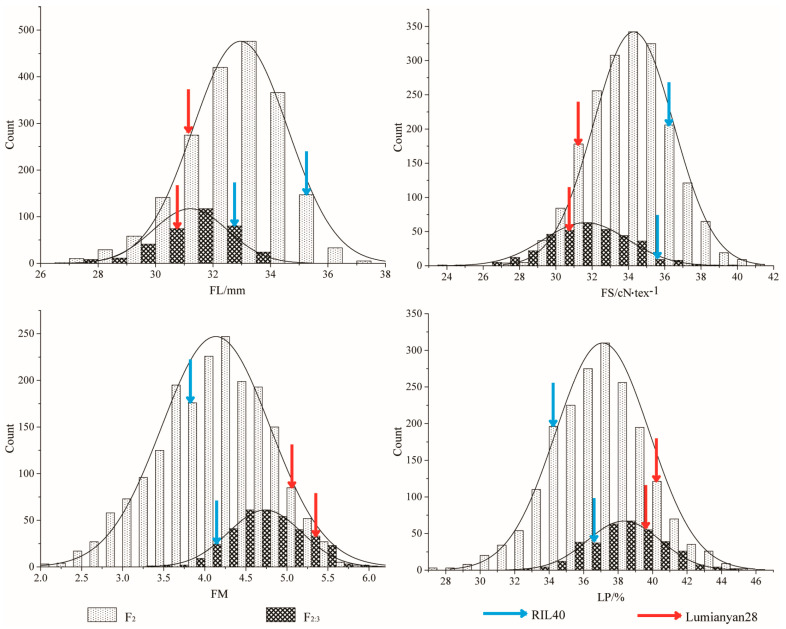
The phenotypic distribution of fiber quality traits and lint percentage of F_2_ and F_2:3_ populations. Dotted diamond bar presents phenotype distribution of F_2_ population; Trellis presents phenotype distribution of F_2:3_ population. The curve on the graph represents the fitted normal distribution of the population. The red and blue arrows indicate the positions of L28 and RIL40 in the distribution, respectively. FL, fiber length; FS, fiber strength; FM, fiber mocronaire; LP, lint percentage.

**Figure 2 plants-12-03737-f002:**
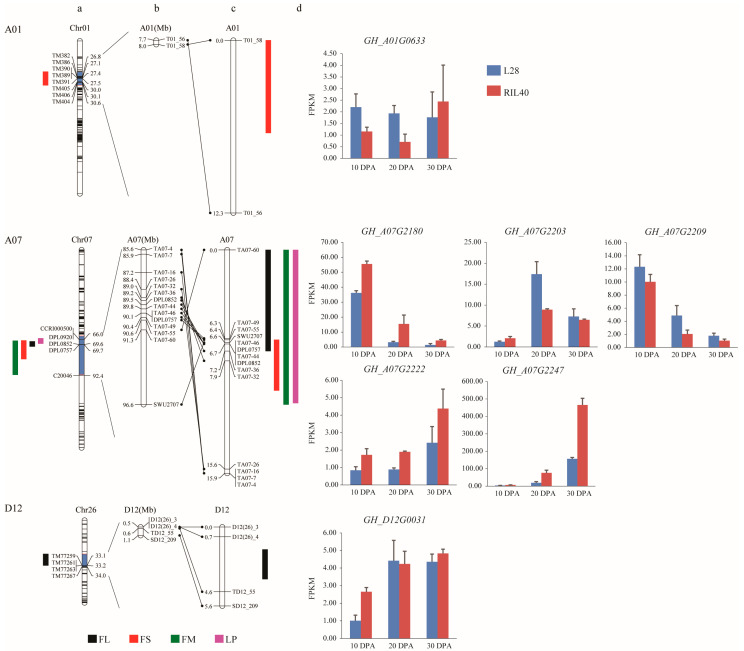
QTLs of fiber quality traits and LP on genetic linkage maps. (**a**) QTLs of fiber quality traits (FL, FS, and FM) and LP on A01, A07, and D12 in primary linkage analysis of RIL population; (**b**,**c**) QTLs of fiber quality traits (FL, FS, and FM) and LP on the physical maps, and on the linkage maps of secondary segregating population; (**d**) differentially expressed genes (DEGs) in developing fibers between L28 and RIL40 in the target QTL intervals.

**Figure 3 plants-12-03737-f003:**
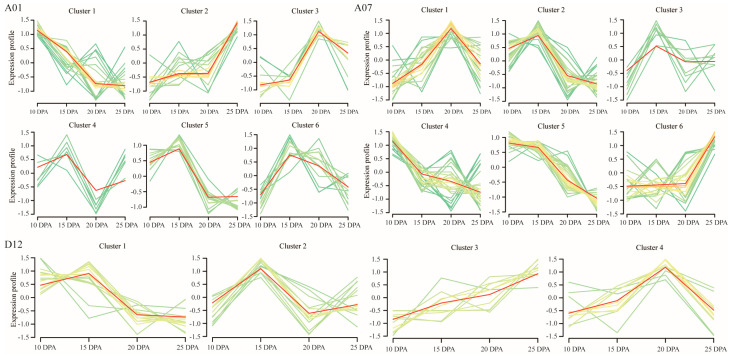
Expression clustering of the dynamically expressed genes in the QTL intervals on chromosomes A01 (6.70–10.15 Mb), A07 (85.48–93.43 Mb), and D12 (0.40–1.43 Mb) in TM-1 gene expression database. Each cluster presents a similar gene expression profiling. The red zigzag line in the figure presents the fitted expression trend of each gene cluster. The yellow lines represent the gene expression profiles are more proximal to the fitted expression trend, while the green lines less proximal to the fitted expression trend.

**Figure 4 plants-12-03737-f004:**
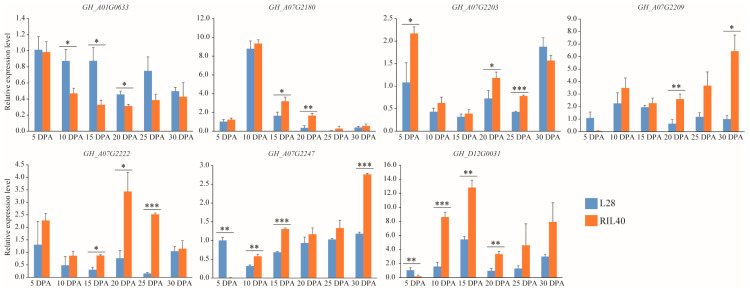
qRT-PCR analysis of candidate genes during fiber development of L28 and RIL40. *, ** and *** indicate the difference between L28 and RIL40 reaching a significant level at *p* < 0.05, *p* < 0.01 and *p* < 0.001 in *t*-test, respectively.

**Table 1 plants-12-03737-t001:** Basic information of the QTL clusters in the primary QTL analysis in a previous study [16].

Chromosome	QTL Compositions				Physical Position (Mb)
	FS	FL	FM	LP	
A01	qFS-chr01-2				7.63–8.04
A07	qFS-chr07-2	qFL-chr07-2	qFM-chr07-1	qLP-chr07-3	89.53–90.08
D12		qFL-chr26-1			0.46–0.52

**Table 2 plants-12-03737-t002:** Fiber quality traits and LP of parental lines and secondary segregating populations of F_2_ and F_2:3_.

		Parent Materials			Population							
Trait	Year	L28	RIL40	|AVDP|	Generation	Range	Min	Max	Average	SD	Skewness	Kurtosis
FL/mm	2019	31.39	35.44 **	4.06	F_2_	10.35	27.22	37.57	32.95	1.65	−0.42	0.15
	2020	30.90	32.94 *	2.04	F_2:3_	7.37	26.62	33.99	31.21	1.30	−0.65	0.49
FS/cN∙tex^−1^	2019	31.28	36.49 **	5.20	F_2_	14.30	27.02	41.32	34.28	2.21	0.06	−0.13
	2020	30.18	35.05 *	4.87	F_2:3_	15.68	23.61	39.29	31.64	2.34	0.01	0.32
FM	2019	5.05	3.79 **	1.26	F_2_	3.88	2.14	6.02	4.13	0.66	−0.25	−0.31
	2020	5.37	4.01 *	1.36	F_2:3_	2.58	3.35	5.93	4.73	0.45	−0.02	−0.22
LP/%	2019	40.21	34.20 **	6.01	F_2_	19.20	27.45	46.65	37.10	2.72	−0.05	0.28
	2020	39.36	36.16 *	3.20	F_2:3_	11.86	32.97	44.83	38.34	2.12	0.09	−0.14

* and ** indicate that the difference between L28 and RIL40 reaches significance levels of *p* < 0.05 and *p* < 0.01, respectively, in *t*-test. |AVDP| represents absolute values of difference between L28 and RIL40.

**Table 3 plants-12-03737-t003:** Correlation analysis of fiber quality traits and LP of secondary segregating populations of F_2_ and F_2:3_.

Trait	Generation	FL/mm		FS/cN∙tex^−1^		FM		LP/%	
		F_2_	F_2:3_	F_2_	F_2:3_	F_2_	F_2:3_	F_2_	F_2:3_
FL/mm	F_2_	1							
	F_2:3_	0.593 **	1						
FS/cN∙tex^−1^	F_2_	0.252 **	-	1					
	F_2:3_	-	0.300 **	0.328 **	1				
FM	F_2_	−0.348 **	-	−0.028	-	1			
	F_2:3_	-	−0.434**	-	0.163 **	0.594 **	1		
LP/%	F_2_	−0.351 **	-	−0.189 **	-	0.508 **	-	1	
	F_2:3_	-	−0.453^**^	-	−0.008	-	0.597 **	0.583 **	1

** indicate the correlation significances between different traits at the 0.01 levels.

**Table 4 plants-12-03737-t004:** QTL verification results of fiber quality traits and LP on A01, A07, and D12 in secondary segregating populations of F_2_ and F_2:3_.

Chromosome	Trait	QTL	Generation	Position (cM)	Marker Interval	LOD	Additive	Dominant	R^2^/%	Physical Interval
A01	FS	qFS-A01-1	F_2:3_	1.01	T01_58-T01_56	3.40	0.10	−1.12	1.45	7.73–8.00
A07	FL	qFL-A07-1	F_2_	6.31	TA07-49-TA07-36	38.84	0.55	0.47	2.31	88.95–91.29
			F_2:3_	0.01	TA07-60-TA07-32	14.98	0.51	0.70	8.79	
	FS	qFS-A07-1	F_2_	8.01	TA07-49-TA07-32	42.53	0.89	0.34	5.79	88.95–90.63
	FM	qFM-A07-1	F_2_	6.61	TA07-49-TA07-36	71.33	−0.35	−0.09	11.34	88.95–91.29
			F_2:3_	8.91	TA07-60-TA07-32	29.85	−0.29	−0.04	31.87	
	LP	qLP-A07-1	F_2_	6.61	TA07-49-TA07-36	73.40	−1.53	−0.29	12.67	88.95–91.29
			F_2:3_	8.91	TA07-60-TA07-32	26.57	−1.30	−0.40	27.36	
D12	FL	qFL-D12-1	F_2_	2.71	D12(26)_3-TD12_55	4.44	0.17	−1.26	0.03	0.48–0.59

**Table 5 plants-12-03737-t005:** Seven DEGs/candidate genes of between L28 and RIL40 identified via RNA-seq at different fiber development stages.

Gene ID	10 DPA *		20 DPA		30 DPA		Gene Name	*Arabidopsis* ID	ArabDesc
	FDR	Log2FC	FDR	Log2FC	FDR	Log2FC			
*GH_A01G0633*	0.01	−1.05	0.01	−1.51	-	-	*CBSX5*	*AT5G53750*	CBS domain-containing protein
*GH_A07G2180*	-	-	0.00	2.20	-	-	*NA*	*AT3G13130*	transmembrane protein
*GH_A07G2203*	-	-	0.00	−1.01	-	-	*COL9*	*AT3G07650*	CONSTANS-like 9
*GH_A07G2209*	-	-	0.01	−1.29	-	-	*RABB1C*	*AT4G17170*	RAB GTPase homolog B1C
*GH_A07G2222*	-	-	0.00	1.05	-	-	*BIR1*	*AT5G48380*	BAK1-interacting receptor-like kinase 1
*GH_A07G2247*	-	-	0.00	1.94	0.00	1.33	*GHL17*	*AT3G07320*	O-Glycosyl hydrolases family 17 protein
*GH_D12G0031*	0.00	1.27	-	-	-	-	*HT1*	*AT3G22750*	Protein kinase superfamily protein

* DPA: days post anthesis.

## Data Availability

The datasets generated during and/or analyzed during the current study are available from the corresponding author upon reasonable request.

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
