# Peer review of "QTL Verification and Candidate Gene Screening of Fiber Quality and Lint Percentage in the Secondary Segregating Population of Gossypium hirsutum"

_plants, 2023, doi:10.3390/plants12213737_

Round 1

Reviewer 1 Report

Comments and Suggestions for Authors

Summary

Fiber quality and yield traits such as fiber strength, fiber length and micronaire (FS, FL, FM) and lint percentage (LP) are critical traits in cotton research. In a previous study based on the L28xX24, RIL population, researchers identified three important QTL/QTL cluster loci. These loci were found on chromosomes A01, A07, and D12 and were shown to harbor at least one of the traits.

In this study, the authors verified the presence of the QTLs within these three intervals by phenotyping the secondary populations L28xRIL40 F2 and F2:3. Additionally, seven differentially expressed genes (DEGs) were identified as potential candidate genes through RNA-seq analysis. Further investigation revealed that five of these genes are likely involved in regulating fiber development, which ultimately leads to the formation of FL, FS, FM, and LP.

Comment

The article has clear objectives and a well-organized experimental design, and the experimental results have achieved the expected effects.

Author Response

 Summary

Fiber quality and yield traits such as fiber strength, fiber length and micronaire (FS, FL, FM) and lint percentage (LP) are critical traits in cotton research. In a previous study based on the L28xX24, RIL population, researchers identified three important QTL/QTL cluster loci. These loci were found on chromosomes A01, A07, and D12 and were shown to harbor at least one of the traits.

In this study, the authors verified the presence of the QTLs within these three intervals by phenotyping the secondary populations L28xRIL40 F2 and F2:3. Additionally, seven differentially expressed genes (DEGs) were identified as potential candidate genes through RNA-seq analysis. Further investigation revealed that five of these genes are likely involved in regulating fiber development, which ultimately leads to the formation of FL, FS, FM, and LP.

Comment

The article has clear objectives and a well-organized experimental design, and the experimental results have achieved the expected effects.

Reply: Thank you very much for taking your precious time reviewing our manuscript and your recognition of our work.

Reviewer 2 Report

Comments and Suggestions for Authors

QTL identification and candidate gene validation offer an alternate tool to discover the genetic and molecular basis of yield and quality traits in cotton crop. Fiber quality traits such as fiber strength, length and micronaire (FS, FL and FM) are important for cotton-based textile industry. However, these molecular and genetic mechanisms are not well studied in the past. Here, authors tried to uncover important QTL/QTL cluster loci, harboring at least one of the above traits on chromosomes A01, A07, and D12 via a recombinant inbred line (RIL) population derived from a cross of Lumianyan28 (L28)×Xinluzao24 (X24).

This manuscript provides experimental basis for need of searching of these functional genes to dissect the genetic mechanism of cotton fiber development. The MS is well described and scientifically sound. It will need some modifications prior to full consideration in this journal.

Some comments:

1.       Authors are requested provide details of molecular methods used for QTL analysis in abstract.

2.       Moreover, the authors are suggested to include the statistical information in abstract and conclusions.

3.       Provide key implications of this study on cotton cultivation and textile industry in China and around the globe.

4.       Recent relevant studies on cotton using DNA based molecular marker analysis for cotton improvement against biotic stresses could be discussed in the introduction. See example, article: Multiplex molecular marker-assisted analysis of significant pathogens of cotton (Gossypium sp.) Chavhan et al., Biocatalysis and Agricultural Biotechnology, 2023

5.       Figure 1 and 3. Caption needs to be elaborated, its too short and don’t describe figure in detail.

6.       Moreover, Figure 1 the F2 highlighted area in graph is not clearly visible. A dark color could be used to visualize.

7.        

Comments on the Quality of English Language

English needs some attention.

Author Response

Comments and Suggestions for Authors

QTL identification and candidate gene validation offer an alternate tool to discover the genetic and molecular basis of yield and quality traits in cotton crop. Fiber quality traits such as fiber strength, length and micronaire (FS, FL and FM) are important for cotton-based textile industry. However, these molecular and genetic mechanisms are not well studied in the past. Here, authors tried to uncover important QTL/QTL cluster loci, harboring at least one of the above traits on chromosomes A01, A07, and D12 via a recombinant inbred line (RIL) population derived from a cross of Lumianyan28 (L28)×Xinluzao24 (X24).

This manuscript provides experimental basis for need of searching of these functional genes to dissect the genetic mechanism of cotton fiber development. The MS is well described and scientifically sound. It will need some modifications prior to full consideration in this journal.

Some comments: 

  1. Authors are requested provide details of molecular methods used for QTL analysis in abstract.

Reply: The method of QTL analysis has been supplemented following your precious comment, please refer to lines 29-30. Thank you very much for your suggestions on abstract section.

  1. Moreover, the authors are suggested to include the statistical information in abstract and conclusions.

Reply: The statistical information in abstract and conclusions has been supplemented according to your suggestions, please refer to lines 27-28 in abstract and lines 404-415 in conclusions. Thank you very much for your suggestions on abstract and conclusions section. 

  1. Provide key implications of this study on cotton cultivation and textile industry in China and around the globe.

Reply: The manuscript has been revised in some discussions of the implications of our study on cotton production and tetile industry, please refer to lines 96-101 in introduction. Thank you greatly for your valuable suggestions.

  1. Recent relevant studies on cotton using DNA based molecular marker analysis for cotton improvement against biotic stresses could be discussed in the introduction. See example, article: Multiplex molecular marker-assisted analysis of significant pathogens of cotton (Gossypiumsp.) Chavhan et al., Biocatalysis and Agricultural Biotechnology, 2023.

Reply: We adopted and added your precious suggestions in the introduction, please refer to lines 65-67. Thank you very much for your suggestions.

  1. Figure 1 and 3. Caption needs to be elaborated, its too short and don’t describe figure in detail.

Reply: The figure captions of Figure 1 and 3 have been detailed following your precious comments, please refer to lines 218-223 and 295-299. Thank you very much for your valuable suggestions.    

  1. Moreover, Figure 1 the F2highlighted area in graph is not clearly visible. A dark color could be used to visualize.

Reply: Thank you very much for your suggestion, we increased the dpi from 300 to 400 of Figure 1 and it seems better in viewing the highlighted area of F2 in it. Thank you very much for your suggestions to make the figure clearer.

Reviewer 3 Report

Comments and Suggestions for Authors

The authors present a study focused on fiber quality traits in cotton, specifically FS, FL, FM, and LP. The researchers aim to understand the genetic basis of these traits. They've identified significant QTLs on chromosomes A01, A07, and D12 using a recombinant inbred line (RIL) population. A secondary segregating population was developed for validation. By performing differential expression analysis on the RNA-seq dataset, the authors obtained several candidate genes that are potentially important. This study provides a good overview of the cotton research and its significance in advancing our understanding of cotton fiber quality traits. However, insufficient and improper analytical strategies may affect important conclusions in the manuscript and make me wonder how reliable the results are.

Major concerns:

1.       The bioinformatics analysis on the RNA-seq dataset should be clearer and more accurate,

a.       Line 229-231, what’s the difference between 571 total genes and 261 ‘dynamic expression’ genes? How to determine if a gene is dynamically expressed during fiber development?

b.       Can authors specify which clusters (in Fig 3) of the genes in Fig 2d come from?

c.       What’s the significance of GO and KEGG pathway enrichment analysis on all expressed genes on three chromosomes? The authors have clustered the genes into different patterns before, don’t the authors want to see the functional differences between these patterns?

d.       Line 263-276, what is the design of differential expression analysis? Is it performed between L28 and RIL40 at each time point? If yes, how do you know which gene is differentially expressed during development and which stage is the highest?  When we want to describe which gene is highly expressed in an indicated time point, we should show a significant p-value compared with other time points rather than different samples at the same time point. The same for qRT-PCR.

2.       The bioinformatics analysis methods performed in this study are not detailed described.

a.       Line 149-151, how can ggplot2 perform GO and KEGG analysis?  The ggplot2 is an R package for the visualization of your enriched terms.

b.       The processing steps of RNA-seq are not detailed enough. For example, what are the QC tools? Adaptor removing tools? Which parameters are used in the mapping and quantification steps? I don’t think the Hisat2 and StringTie can get an “expression pattern”.

c.       The FPKM is calculated with all genes or only the genes on A01. A07, and D12?

d.       What does the “Expression change” mean in Fig 3? Is z scaled FPKM? What do the red and green lines represent in the figure?

e.       I prefer the authors use the dot plot rather than the bar plot in Figure 2d and Fig 4.

f.        In Fig S1-3, there is no need to show all the enriched terms including BP, CC, and MF. Besides, the enriched GO terms with only one or two genes showed significant p-values which surprised me. I hope the authors can upload the code used for all the analysis and visualization in this manuscript.

Minor:

1.       Give the full name of the SSR in line 26.

2.       In Fig2, there are duplicated a-d, which is not allowed in one figure.

3.       No y-labels in Fig.4.

4.       Fig 2d should be cited right after fig 2c, not after Fig 3.

5.       Please add the version of every software, database, and package used in this study.

6.       I’m not sure about the requirements for PLANTs about data availability. Data upon request is not allowed in most journals. The codes for data analysis also need to be uploaded.

Comments on the Quality of English Language

The manuscript needs to be improved.

Author Response

Comments and Suggestions for Authors

The authors present a study focused on fiber quality traits in cotton, specifically FS, FL, FM, and LP. The researchers aim to understand the genetic basis of these traits. They've identified significant QTLs on chromosomes A01, A07, and D12 using a recombinant inbred line (RIL) population. A secondary segregating population was developed for validation. By performing differential expression analysis on the RNA-seq dataset, the authors obtained several candidate genes that are potentially important. This study provides a good overview of the cotton research and its significance in advancing our understanding of cotton fiber quality traits. However, insufficient and improper analytical strategies may affect important conclusions in the manuscript and make me wonder how reliable the results are.

Major concerns:

  1. The bioinformatics analysis on the RNA-seq dataset should be clearer and more accurate,
  2. Line 229-231, what’s the difference between 571 total genes and 261 ‘dynamic expression’ genes? How to determine if a gene is dynamically expressed during fiber development?

Reply: 571 genes are all genes of in physical interval of three QTL clusters, and 261 genes are expressed genes obtained from these 571 genes based on mean FPKM values of each gene at 10, 15, 20, and 25 DPA that are greater than 0.5 in TM-1 RNA-seq data. FPKM value is currently a widely used algorithm for statistical gene expression in RNA-seq analysis. The gene expression clustering trend graphs were drawn using Mfuzz R package. The Mfuzz R package can analyze the temporal dynamic characteristics of gene or protein expression profiles. This tool can identify potential time series patterns of expression profiles and cluster genes with similar patterns to help us understand the dynamic patterns of genes. The description of the section has been revised, please see lines 267-284 in the revised manuscript. Thank you very much for your great suggestions.

  1. Can authors specify which clusters (in Fig 3) of the genes in Fig 2d come from?

Reply: We do the expression profiling analysis (Figure 3) and differential expression (Figure 2d) independently, and then compare their results. The revision of manuscript has clarified your concerns accordingly, please refer to lines 275-282. Thank you very much for your suggestions.

  1. What’s the significance of GO and KEGG pathway enrichment analysis on all expressed genes on three chromosomes? The authors have clustered the genes into different patterns before, don’t the authors want to see the functional differences between these patterns?

Reply: We agree to your valuable comments and the content of GO and KEGG analysis has been removed from the manuscript. Thank you very much for your great suggestions.

  1. Line 263-276, what is the design of differential expression analysis? Is it performed between L28 and RIL40 at each time point? If yes, how do you know which gene is differentially expressed during development and which stage is the highest? When we want to describe which gene is highly expressed in an indicated time point, we should show a significant p-value compared with other time points rather than different samples at the same time point. The same for qRT-PCR.

Reply: 1. Differential expression analysis is based on the Count number of genes in RNA-seq between samples, using differential analysis software DESeq2. Fold Change (FC) represents the ratio of expression levels between two samples (groups). The False Discovery Rate (FDR) is obtained by correcting the p_value of the difference in significance, representing the significance of the difference. For the convenience of comparison, the logarithmic value of FC was taken and expressed as log2FC. The larger the absolute value of log2FC, the more significant the difference between the two groups of samples at significant level of FDR < 0.01. 2. The reason for the analysis of the differential expression genes at the same time point in this study is that there are significant differences between the two parental materials in fiber quality and yield traits. The purpose of statistical analysis of differential expression genes at the same time point between two materials is to find out which differential expression genes at the same time point in these intervals may lead to the differences in fiber quality and yield. Thank you very much for your great comments.

  1. The bioinformatics analysis methods performed in this study are not detailed described.
  2. Line 149-151, how can ggplot2 perform GO and KEGG analysis? The ggplot2 is an R package for the visualization of your enriched terms.

Reply: The content of GO and KEGG analysis has been removed from the article. Thank you for your suggestions.

  1. The processing steps of RNA-seq are not detailed enough. For example, what are the QC tools? Adaptor removing tools? Which parameters are used in the mapping and quantification steps? I don’t think the Hisat2 and StringTie can get an “expression pattern”.

Reply: The detailed analysis steps have been revised and supplemented, as shown in lines 165-179. Thank you very much for point out the errors in our manuscript.

  1. The FPKM is calculated with all genes or only the genes on A01. A07, and D12?

Reply: RNA-seq was analyzed on a genome-wide basis for all genes, but this study only focused on genes in three specific intervals. Therefore, only genes in these specific intervals were described. Thank you very much for your valuable comments.

d.What does the “Expression change” mean in Fig 3? Is z scaled FPKM? What do the red and green lines represent in the figure?

Reply: “Expression change” in Figure 3 was modified as “expression profile”, please refer to lines 294-299. The FPKM values of the genes were analyzed and visualized to obtain the expression profiles using R package Mfuzz. Each cluster presents a similar gene expression profiling. The red zigzag line in the figure presents the fitted expression trend of each gene cluster, while the other colored lines represent the change in expression trend of each gene. Thank you very much for your priceless suggestions.

  1. I prefer the authors use the dot plot rather than the bar plot in Fig 2d and Fig 4.

Reply: Thank you very much for your valuable suggestions. We tried to present the results of Figure 2d and Figure 4 in dot plot, however, we believe that bar plot presented better viewing. On the one hand, it focuses on the expression profiling during fiber development of the two materials and the expression differences between the two materials at the same time point of fiber development. Therefore, the bar plots may present a better visualization of the differences between the two materials.

  1. In Fig S1-3, there is no need to show all the enriched terms including BP, CC, and MF. Besides, the enriched GO terms with only one or two genes showed significant p-values which surprised me. I hope the authors can upload the code used for all the analysis and visualization in this manuscript.

Reply: The content of GO and KEGG analysis has been removed from the article. Thank you very much for your priceless suggestions.

Minor:

  1. Give the full name of the SSR in line 26.

Reply: The full name of the SSR has been added in the manuscript, please refer to line 28. Thank you greatly for your suggestions.

  1. In Fig2, there are duplicated a-d, which is not allowed in one figure.

Reply: The duplications of a-d have been removed from Figure 2, please refer to line 247. Thank you very much for your suggestions.

  1. No y-labels in Fig.4.

Reply: The y-axis has in Figure 4 has been labeled, please refer to line 300. Thank you very much for your valuable suggestions.

  1. Fig 2d should be cited right after fig 2c, not after Fig 3.

Reply: Citation of Figure 2c and Figure 2d has been readjusted, please refer to lines 229-284. Thank you very much for your suggestions.

  1. Please add the version of every software, database, and package used in this study.

Reply: The versions of software, database, and package were supplemented in the relevant places of materials and methods section, please refer to lines 163-184. Thank you very much for your suggestions.

  1. I’m not sure about the requirements for PLANTs about data availability. Data upon request is not allowed in most journals. The codes for data analysis also need to be uploaded.

Reply: In this study, some analysis was based on publicly available online website platform and does not have any code to upload. The transcriptome data of the two parental lines (L28 and RIL40) in 10, 20, and 30 DPA were used from unpublished data, which will be made available on line in whole data set with another manuscript. Thank you very much for your great comments.

Round 2

Reviewer 3 Report

Comments and Suggestions for Authors

All the concerns have been revised.